# Population Genetics of *Phlebotomus papatasi* from Endemic and Nonendemic Areas for Zoonotic Cutaneous Leishmaniasis in Morocco, as Revealed by *Cytochrome Oxidase* Gene Subunit I Sequencing

**DOI:** 10.3390/microorganisms8071010

**Published:** 2020-07-06

**Authors:** Souad Guernaoui, Omar Hamarsheh, Deborah Garcia, Didier Fontenille, Denis Sereno

**Affiliations:** 1Laboratory of Biotechnology and Valorisation of Plant Genetic Resources, Faculty of Sciences and Techniques, University of Sultan Moulay Slimane, Béni Mellal B. P 523, Morocco; 2IRD, Université de Montpellier, MIVEGEC, 34000 Montpellier, France; deborah.garcia@ird.fr (D.G.); didier.fontenille@ird.fr (D.F.); 3Sidi Mohamed Ben Abdellah University, Fes 30000, Morocco; 4Department of Biological Sciences, Al-Quds University, Jerusalem P.O. BOX 51000, Palestine; ohamarsheh@gmail.com; 5IRD, Université de Montpellier, InterTryp, 34 000 Montpellier, France

**Keywords:** *Phlebotomus papatasi*, *Leishmania major*, zoonotic cutaneous leishmaniasis, genetic structure, *Cytochrome oxidase subunit I*, Morocco

## Abstract

Zoonotic cutaneous leishmaniasis (ZCL) caused by *Leishmania major* Yakimoff & Shokhor and transmitted by *Phlebotomus papatasi* (Scopoli) is a public health concern in Morocco. The disease is endemic mainly in pre-Saharan regions on the southern slope of the High Atlas Mountains. The northern slope of the High Atlas Mountains and the arid plains of central Morocco remain non-endemic and are currently considered high risk for ZCL. Here we investigate and compare the population genetic structure of *P. papatasi* populations sampled in various habitats in historical foci and non-endemic ZCL areas. A fragment of the mtDNA *cytochrome oxidase I (COI)* gene was amplified and sequenced in 59 individuals from 10 *P. papatasi* populations. Haplotype diversity was probed, a median-joining network was generated (*F*_ST_) and molecular variance (AMOVA) were analyzed. Overall, we identified 28 haplotypes with 32 distinct segregating sites, of which seven are parsimony informative. The rate of private haplotypes was high; 20 haplotypes (71.4%) are private ones and exclusive to a single population. The phylogenetic tree and the network reconstructed highlight a genetic structuration of these populations in two well defined groups: Ouarzazate (or endemic areas) and Non-Ouarzazate (or nonendemic areas). These groups are separated by the High Atlas Mountains. Overall, our study highlights differences in terms of population genetics between ZCL endemic and non-endemic areas. To what extent such differences would impact the transmission of *L. major* by natural *P. papatasi* population remains to be investigated.

## 1. Introduction

The Leishmaniases are a group of parasitic diseases caused by various species of protozoa belonging to the genus Leishmania (Trypanosomatida, Kinetoplastidae) and transmitted by several species of phlebotomine sand fly vectors (Diptera, Psychodidae) [1,2,3]. *Phlebotomus papatasi* (Scopoli, 1786) is the proven vector of *L. major* Yakimoff & Shokhor 1914, which causes zoonotic cutaneous leishmaniasis (ZCL) in North Africa and other geographic areas, with rodents acting as reservoir hosts [4,5]. In Morocco, *L. major* zymodeme MON 25 is transmitted by *P. papatasi* [6,7], and *Meriones shawi grandis* is the reservoir host [8]. The first registered historical outbreak of ZCL was in the 1970s throughout the south of the High Atlas Mountains, from Ouarzazate to Errachidia (Figure 1). Data from the health ministry [9] pinpoint that ZCL is increasing but still remains mainly subordinated to the pre-Saharan zone on the southern slope of the High Atlas Mountains, despite the presence of the proven vector and reservoir for *L. major* as well as favorable eco-climatic conditions. These areas should therefore be classified as at high risk for ZCL [10,11].

*Phlebotomus papatasi* present a wide geographical distribution [12] and adapt to diverse habitats including highly disturbed ones [13,14,15,16]. Strikingly, the distribution of *P. papatasi* exceeds that of *L. major* and its reservoir hosts, extending to southern Europe and the eastern regions of the Indian subcontinent, where it is often abundant in domestic and peri-domestic biotopes far from rodent colonies [4,17]. It is hypothesized that two cryptic populations are involved in the transmission of *L. major* between rodents and from rodents to humans [17]. The underlying population structure and genetic variability within and among geographically distant populations may influence vectoral capacity, thus having epidemiological implications warranting assessment of control strategies to prevent transmission of CL. Overall population genetics analysis with *cytochrome b* mtDNA sequences showed that *Phlebotomus papatasi* populations are relatively homogeneous [18] despite pockets of genetic variation [19,20,21,22,23]. Such molecular homogeneity in diverse geographical populations of *P. papatasi* is also demonstrated with another molecular marker, the second internal transcribed spacer ribosomal DNA (rDNA *ITS2*) [24]. Microsatellite markers applied to *P. papatasi* populations from the North African and the Mediterranean subregions have provided evidence for genetic differentiation [25,26] and have also revealed genetic differentiation in Sudanese *P. papatasi* populations at a microgeographic scale [27].

In Morocco, *P. papatasi* are one of the most frequently trapped sandfly species [12,28,29,30]; they express a preference for arid and perarid bioclimates [29] and their distribution is limited by elevation [31]. High densities of *P. papatasi* are recorded in urban areas and in unhealthy suburbs of large cities such as Marrakech and Agadir, particularly in wild dumps and areas of sewage application, far away from human dwellings [16]. Phenotypic variations or “abnormalities” in the male genitalia of *P. papatasi* are frequently recorded, with symmetrical anomalies being observed in the lateral lobes, along with abnormal styles, which correlate with environmental human-caused disturbances [15]. Here, using the *cytochrome c oxidase I* gene as a molecular marker, we investigated the genetic structure of wild *P. papatasi* populations sampled in endemic and non-endemic areas for ZCL in Morocco.

## 2. Materials and Methods

### 2.1. Origins and Sampling of Sand Flies

Sand fly specimens were collected between April and November from 2006 to 2008. They were captured using CDC-miniature light traps placed in houses and sticky traps (castor oil papers) placed in intra-domestic, peri-domestic, and sylvatic habitats (Table 1). Collected sand flies were stored in 96% ethanol. Sampling was performed in two geographic areas (Figure 1). The first one corresponded to the historical focus of ZCL due to *L. major* in Ouarzazate, southern Morocco. The second one, in central and southwestern Morocco, is a non-endemic area for *L. major*. The two areas are separated by the High Atlas Mountains (Figure 1).

### 2.2. Morphological Identification

The head and the abdominal Terminalia of each specimen were dissected with sterilized microneedles and mounted on a slide in Canada balsam [32]. Morphological identification at the species level was performed by the examination of the external genitalia and pharyngeal armatures for males. The rest of the body was used for DNA extraction.

### 2.3. DNA Extraction, Cytochrome Oxidase Subunit I (COI) Amplification, and DNA Sequencing

Total DNA was extracted using the QIAmp kit (Qiagen, Manchester, England), according to the manufacturer’s protocol.

Polymerase chain reaction (PCR) was performed in a total volume of 50 µL. The primers LepF (5′-ATTCAACCAATCATAAAGATATTGG-3′) and LepR (5′-TAAACTTCTGGATGTCCAAAAAATCA-3′) were used to amplify a 681 bp fragment of the *COI* gene [33]. Each amplification reaction contained 4 µL of DNA, each primer at 0.5 µM final concentration, 0.2 mM dNTPs and 0.02 U of Phusion DNA Polymerase (Finnzyme). Amplification was performed according to the following cycling conditions: an initial denaturation step at 98 °C for 45 s; followed by 35 cycles at 98 °C for 10 s, 57 °C for 30 s and 72 °C for 30 s; and a final extension at 72 °C for 7 min. The PCR products were directly sequenced in both directions using the primers used for DNA amplification.

### 2.4. Data Analysis

Sequences (681 bp) were manually edited and aligned with the online service Multalin [34] (http://ribosome.toulouse.inra.fr/multalin/multalin) using the *cytochrome c oxidase I* gene as a molecular marker. Gaps were treated as missing data. Sequence analysis and Kimura’s two-parameter genetic distance calculation were performed using MEGA v. 4.0 [35,36]. The estimation of nucleotide diversity between and within populations using the Φ [37,38] and π statistics [39] was performed with ARLEQUIN v. 3.0 [40]. ARLEQUIN v. 3.0 was also used for calculating Tajima’s D and Fu’s Fs, and molecular analysis of variance (AMOVA) was used to determine the genetic structure of populations and their differentiation (*F*_ST_) [39]. Significance was tested by performing 1000 permutations. *F*st values > 0.25 indicate strong genetic differentiation. The median-joining algorithm, implemented in the software package NETWORK v. 3.1 [41,42], was used to construct a median-joining network. Phylogenetic analysis was carried out using maximum likelihood analysis, using the default settings of the online phylogeny website (https://www.phylogeny.fr/) [43]. Alignments were done on the 654 bp of *P. papatasi COI* gene sequences from Morocco, or from Spain (LC090048.1), Serbia (KY848828.1), Turkey (MH780862.1), Cyprus (MF968974.1), and of *P. ariasi* (KJ1481169.1), *P. sergenti* (KC755398.1), or *P. perniciosus* (KJ1481136.1).

## 3. Results

### 3.1. Sequence Analysis

The PCR amplification of the 3′ end of the mitochondrial *COI* gene produced a band of approximately 700 bp in a 1% agarose gel. The 681 bp sequences were manually trimmed, resulting in good quality sequences with a length of 654 bp. Sequence alignment of the 654 bp *COI* sequences did not identify any nucleotide insertion/deletions (indels), and sequences can be fully translated with the Drosophila mitochondrial code. Seven parsimony-informative sites were recorded in more than one haplotype. All sequences are available in GenBank under accession numbers MT074053-054, MT075056-57, MT075059-61, MT075064-75, MT075057, and MT075078-85. Sequence analysis delineated the presence of 28 haplotypes with 32 distinct segregating sites (Table 2). A relatively low abundance of GC and an excess of AT (59.98%) is recorded. Twenty-nine substitutions are identified, including 17 A→G transitions and 12 T→C transversions; transitions outnumbered transversions. The average haplotype diversity (0.9381 +/− S.D = 0.0403) and nucleotide diversity (θ*_S_* = 5.41756 +/− 0.59318; θ*π* = 4.86013 +/− 0.20454; π = 486,013 +/− 0.20454) are high. Finally, Tajima’s D and Fu’s Fs neutrality values are negative and significant in all sampled populations. These pinpoint probable episodes of population expansion (mean Tajima’s D = −0.36383, *p* = 0.39850; mean Fu’s Fs = −3.57473, *p* = 0.18700). See Appendix A for more information on statistics (Appendix A).

### 3.2. Haplotype Diversity and Distribution

The 28 haplotypes recorded within the 59 *P. papatasi* samples are presented in Table 2. Haplotype H8 is the most frequently recorded. It is present in seven populations from non-endemic areas. Other haplotypes are less frequent; H1 and H25 are present in three populations, and H14 is recorded in only two populations (Table 3). The haplotype H25 is the only shared haplotype between *P. papatasi* sampled in endemic and non-endemic areas. The other 24 of the 28 identified haplotypes (85.7%) are private ones. Thirty private haplotypes (H10–H13, H15, H17–H19, H22–H24, H26–H27) are associated with Ouarzazate, five (H2–H4, H7, H21) with Nfifa, one (H9) with Labrouj, and one with Taferiat (H5). The rate of private haplotypes is high in populations from the ZCL-endemic areas of Ouarzazate. Here, 13 of the 15 identified haplotypes (86.6%) are unique. In non-endemic areas, the corresponding rate is only of 42.8% (6 out of 14 haplotypes).

### 3.3. Phylogenetic Analysis

*Phlebotomus papatasi* belongs to the Phlebotomus subgenus. The consensus tree (Figure 2A) recovered displays that samples from Morocco fall within the clade of other *P. papatasi* samples collected in Turkey, Spain and Cyprus. They are clearly divergent from *P. (Laroussius) ariasi*, *P. (Laroussius) perniciosus* or *P. (Transphlebotomus) sergenti* which belong to the *Laroussius* and *Transphebotomus* subgenus. Overall, within *P. papatasi* samples collected in Morocco, the presence of at least two *P. papatasi* populations that fall into two groups which show some genetic relatedness is depicted (bootstrap support values above 50%). Most specimens sampled in the endemic area of Ouarzazate are grouped together and constitute the “Ouarzazate group”, with a bootstrap value of 56%. This group is interrupted by two individuals from Labrouj in the central Morocco area. The other specimens from non-endemic areas, or “Non-Ouarzazate” areas, constitute a geographically mixed group.

### 3.4. Median-Joining Network Analysis

The median-joining analysis displays a highly structured network. Within each major clade, the internal and terminal nodes can be interpreted as old and recently derived haplotypes. The network display places all haplotypes into two groups: “Ouarzazate” or endemic; and “Non-Ouarzazate” or non-endemic. Within the Non-Ouarzazate group, two groups (named Mix1 and Mix2) are distinguishable (Figure 3). These two groups encompass haplotypes from the Ouarzazate and Non-Ouarzazate areas. The Mix2 group is closer to the Ouarzazate group in term of haplotype composition than Mix1—it might be considered as a sister group of the latter. Overall, the clustering displayed by the median-joining network agrees well with the topology of the trees generated by the maximum likelihood analysis. The clear variation in clustering between haplotypes of endemic and non-endemic supports a genetic differentiation between these populations.

### 3.5. Population Differentiation

The genetic differentiation among populations of *P. papatasi* was defined by *F* statistics. AMOVA results presented in Table 4 confirm variation among and within populations; 31.03% and 68.97%, respectively (Table 4).

The *F*_ST_ values were calculated in a pairwise manner for the main populations (Ouarzazate and Non-Ouarzazate, encompassing Mix 1 and Mix2). With a *F_ST_* value of 0.31026, the matrix of significance for 110 permutations reached *p* = 0.05, which highlights some genetic differentiation between the two groups that were analyzed using our 59 samples.

## 4. Discussion

Natural barriers like high mountains play an important role for limiting dispersal and isolating populations. The Atlas, a natural barrier, may have played a role in limiting the diffusion of zoonotic cutaneous leishmaniasis to the Saharan regions of Morocco. This Atlas consists of three parallel ranges: the Anti-Atlas in the southwest, which gives way to the High Atlas, where the country’s highest peak rises to 4165 m; the Middle Atlas is located northeast of the High Atlas. Rolling plateaus east of the mountains gradually lead into the Sahara, located in southeastern and southern Morocco. Here, in sub-Saharan areas, *L. major* is largely widespread. The central Moroccan plateaus overlooking the northern slope of the High Atlas, notably the Haouz plain and Chichaoua, remain non-endemic but, according to entomological data, is considered high risk.

We have investigated the genetic structure of 10 populations collected from both endemic and non-endemic areas for zoonotic cutaneous leishmaniasis using the *cytochrome c oxidase subunit I* (*Cox1* or *COI*) as molecular marker to gather clues on the influence of the Atlas Mountains on the evolutionary history of *P. papatasi* populations in Morocco. The *cytochrome c oxidase* gene is used as a DNA barcode to guide the identification of new animal species and to delineate cryptic taxon and the association between their life stages [44,45,46]. The sequence polymorphism at this locus has been used to investigate the population genetic structure of *Lutzomyia longipalpis* [47], and to discriminate *L. umbratilis* sibling species [48]. In Old World sand fly species, *COI* in combination with the *cytochrome b* (*Cytb*) gene can differentiate two closely related species, *P. chabaudi* and *P. riouxi* [49]. Here, we observe a genetic diversity of the *COI* gene in Morocco, reminiscent of a genetic structuration in *P. papatasi* populations. Samples collected were indistinguishable by morphological observation and genomic analysis of the 18srRNA (data not shown). The haplotype distributions displayed in the median-joining network as well as the phylogenetic analysis supported the notion of a geographic clustering of *P. papatasi*, in Morocco. Two groups, with distinct genetic structures, can be differentiated. The first one encompasses *P. papatasi* samples from the ZCL endemic areas of Ouarzazate. The second one, designated the “Non-Ouarzazate” group, includes a mix of individuals from non-endemic areas sampled on the northern slope of the High Atlas Mountains and the arid plains of central Morocco, as well as samples from endemic areas.

The median-joining network, as well as nucleotide analyses, provide additional evidence of a probable demographic expansion of *P. papatasi* populations. A complex scenario of *P. papatasi* populations might have taken place in Morocco, with expansion events allowing the appearance of multiple haplotype subgroups. The Ouarzazate population of *P. papatasi*, which derives from the populations of the southern slope of the High Atlas Mountains, is expected to have experienced a period of isolation due to the physical barrier of the Atlas Mountains. Then, population expansion and genetic diversification might have occurred, maybe following a climatic shift. These assumptions are supported by the high rate of private haplotypes recorded in this group (87%), which could reflect some local adaptation. The diversity of the ecological conditions may affect the distribution and density of *P. papatasi*, including the bioclimate, anthropization, urbanization, and elevation [50]. The bioclimate is Saharan in Ouarzazate. At stations from central and southwestern Morocco, it varies from arid in Nfifa, Labrouj and Chouiter to semi-arid and sub-humid in Taferiat, Amizmiz, Zinit, Ijjoukak, Sti Fadma and Oukaemiden. In non-endemic areas, sampling was performed on plains where the *P. papatasi* density is high and in mountainous localities where *P. papatasi* is less frequent [31]. Despite the very high diversity in the ecological conditions, samples from non-endemic areas were always genetically intermingled, forming the Non-Ouarzazate group. This group includes Mix1, close to *P. papatasi* from endemic areas, and the more distant Mix2. The Mix1 and Mix2 populations are not geographically isolated by the High Atlas Mountains. It therefore appears that ecological habitats do not play a key role in shaping the genetic variation of *P. papatasi* populations. This finding agrees with other reports from Hamarsheh et al. [22,25,26] that highlight a genetic structure of *P. papatasi* linked to latitude rather than to ecological environment.

It will be now of interest to gather more precise information on the evolutionary scenarios that have governed the dispersal of *P. papatasi* populations and therefore have had an impact on the current and future ZCL incidence in Morocco. To this aim, the use of microsatellite markers may help to more precisely define the demographic history of Moroccoan *P. papatasi* populations.

## 5. Conclusions

This work, based on a maternally inherited marker (the *COI* gene), highlights clear differences in the genetic structure between *P. papatasi* populations living the *L. major*-endemic area of Ouarzazate and those of non-endemic areas of central Morocco. In addition, morphological differences have been reported in *P. papatasi* populations in Morocco [51]. Indeed, these populations sampled from both sides of the High Atlas Mountains also show genetic differences in another marker, the galectin gene [52], which plays a pivotal role in the recognition of *L. major* by its specific vector *P. papatasi* [53]. This study pinpoints a polymorphism in the galectin gene (*PpGalec*) with a mutation affecting an amino acid involved in substrate recognition. Although performed on a limited number of samples, this further highlights an unequal distribution of this mutation in populations from endemic and non-endemic areas for zoonotic cutaneous leishmaniasis [53]. All these reported observations question potential variation in the vectoral capacity/competence of *P. papatasi* populations from the southern and the northern slope of the High Atlas Mountains in Morocco.

## Figures and Tables

**Figure 1 microorganisms-08-01010-f001:**
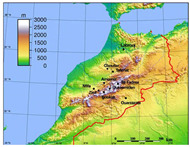
Geographical location of sites where *P. papatasi* were trapped.

**Figure 2 microorganisms-08-01010-f002:**
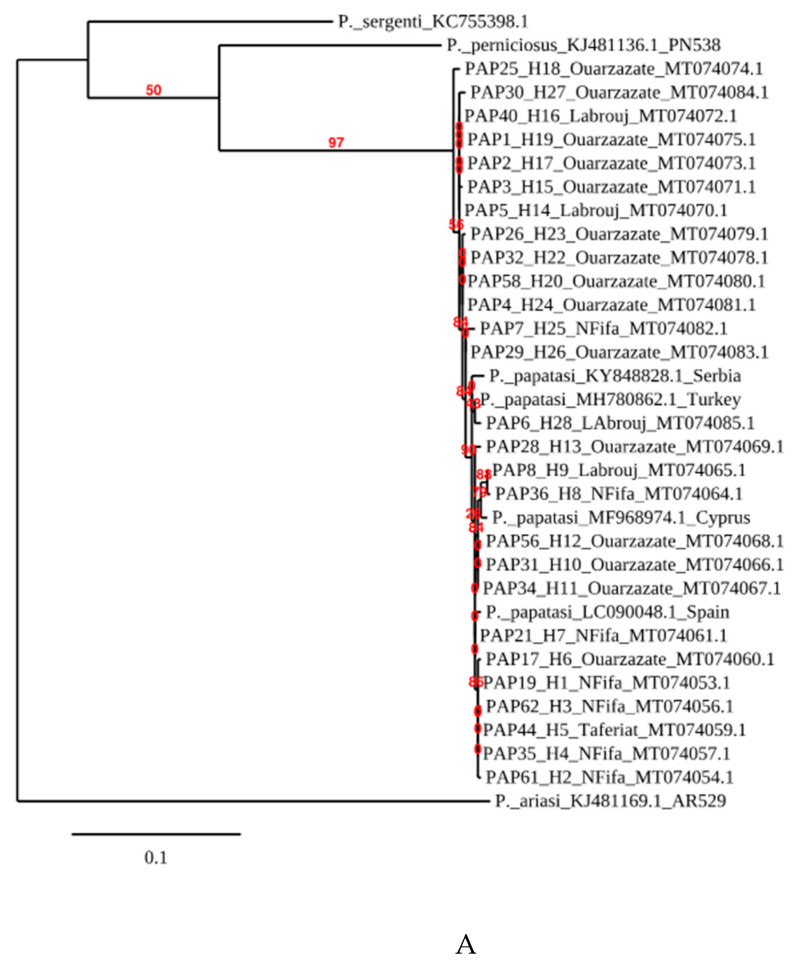
Phylogenetic tree reconstructed from (**A**) *P. papatasi COI* sequences from Morocco, Serbia, Turkey and Spain. *P. ariasi* was used as an outgroup. (**B**) *P. papatasi* samples from Morocco.

**Figure 3 microorganisms-08-01010-f003:**
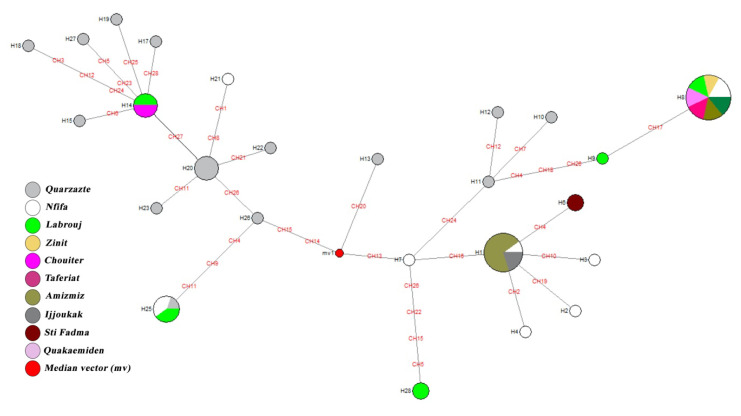
Median-joining network deduced from *P. papatasi COI* sequences sampled in Morocco. Circle size and color are indicative of the frequency and the geographical location of haplotypes. Haplotype identification is provided next to the corresponding circle. Nucleotide substitutions are highlighted in red.

**Table 1 microorganisms-08-01010-t001:** Sampling stations in the study area.

Stations	Latitude	Longitude	Altitude (m)	Number of Specimens	Trapping Method
Ouarzazate	31°23′19 N	5°59′15 W	1648	20	CDC trap
Nfifa	31°32′40 N	8°45′58 W	322	10	CDC trap
Labrouj	32°30′0 N	7°12′0 W	388	6	CDC trap
Chouiter	31°34′0 N	7°49′0 W	529	2	Sticky trap
Taferiat	31°32′59 N	7°36′29 W	755	5	Sticky trap
Amizmiz	31°13′0 N	8°15′0 W	1004	8	Sticky trap
Zinit	31°4′60 N	8°41′60 W	1148	2	CDC trap
Ijjoukak	31°0′0 N	8°10′0 W	1233	2	CDC trap
Sti Fadma	31°13′0 N	7°42′0 W	1772	2	CDC trap
Oukaemiden	31°12′21 N	7°51′51 W	2613	2	CDC trap

**Table 2 microorganisms-08-01010-t002:** *P. papatasi* haplotypes within the 59 Moroccan *P. papatasi* specimens sampled.

Haplotypes	Specimen Code	Nb	Variant Character Position
H1	PAP78,70,43,41, 19,13,11,10,9	9	GGTCGCA CCTTGGGCGA TTAGGAGTAG CTATAT
H2	PAP61	1	....... .......... .C........ ......
H3	PAP62	1	....... ..A....... .......... ....G.
H4	PAP35	1	.A..... .......... .......... ......
H5	PAP44	1	....... .......... .......... ...C..
H6	PAP79,17	2	...T... .......... .......... ......
H7	PAP21	1	....... ........A. .......... ......
H8	PAP80,77,76,75,71,55,47, 46,45,36,22,18,12,9	14	...T... ........AG C.....A.G. ......
H9	PAP8	1	...T... ........A. C.....A.G. ......
H10	PAP31	1	......G ........A. ......A... ......
H11	PAP34	1	....... ........A. ......A... ......
H12	PAP56	1	....... ....A...A. ......A... ......
H13	PAP28	1	....... .....A..A. ..G....... ......
H14	PAP5,49	2	....... .....AATA. ........GA ......
H15	PAP3	1	.....T. .....AATA. ........GA ......
H16	PAP72,40	2	....... .....AATA. ........GA .....C
H17	PAP2	1	....... .....AATA. ........GA T.....
H18	PAP25	1	..C.... ....AAATA. ......A.GA .C....
H19	PAP1	1	....... .....AATA. .......GGA ..G...
H20	PAP58,33,24	3	....... .....AATA. ........G. ..G...
H21	PAP37	1	C...... T....AATA. ........G. ..G...
H22	PAP32	1	....... .....AATA. ...A....G. ..G...
H23	PAP26	1	....... ...C.AATA. ........G. ..G...
H24	PAP4	1	....... .....AATA. ........G. ......
H25	PAP67,60,57,38,7	5	...T... .T.A.AATA. .......... ..G...
H26	PAP29	1	....... .....AATA. .......... ......
H27	PAP30	1	....A.. .....AATA. .....G..GA ......
H28	PAP73,6	2	....A.. .......TA. ....A...T. ......

Alignment of variant nucleotide positions for the 654 bp of the mitochondrial *COI* gene. Only polymorphic positions are shown. Spaces represent identity with respect to the first reference sequence.

**Table 3 microorganisms-08-01010-t003:** Distribution of *COI* haplotypes within the Moroccan populations of *P. papatasi* included in this study. Unique haplotypes are underlined.

Stations	*COI*-Haplotypes	Unique Haplotypes
**Ouarzazate**	H10, H11, H12, H13, H15, H17, H18, H19, H20, H22, H23, H24, H25, H26, H27	13
**Nfifa**	H1, H2, H3, H4, H7, H8, H21, H25	5
**Labrouj**	H8, H9, H14, H16, H25, H28	1
**Chouiter**	H8, H14	0
**Taferiat**	H5, H8	1
**Amizmiz**	H1, H8	0
**Zinit**	H8	0
**Ijjoukak**	H1	0
**Sti Fadma**	H6	0
**Oukaemiden**	H8	0

**Table 4 microorganisms-08-01010-t004:** Molecular variance (AMOVA).

Source of Variation	* d.f	Sum of Squares	Variance Components	Variation (%)
**Among populations**	1	31.716	1.08*341* Va	31.03
**Within populations**	57	137.284	2.40850 Vb	68.97
**Total**	58	169.000	3.49190	/
**Fixation index (*F_ST_*)**	0.31026	/	/	/

* d.f: degree of freedom.

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
