# Peer review of "Population Genetics of Phlebotomus papatasi from Endemic and Nonendemic Areas for Zoonotic Cutaneous Leishmaniasis in Morocco, as Revealed by Cytochrome Oxidase Gene Subunit I Sequencing"

_microorganisms, 2020, doi:10.3390/microorganisms8071010_

Round 1

Reviewer 1 Report

  1. The use of CO1 marker in sand flies is problematic, since it is one of most variable genetic markers that can be used in sand fly barcoding not only in Morocco. This article supports this assertion. The fact that there is a multiple haplotype in each focus weakens the finding, but on the other hand the results and the way the analysis is presented here is valued and important.
  2. It is challenging to determine haplotype based in one sequence only.
  3. There is no other morphological, molecular or biological evidence to support the most important statement in this work "the CO1 gene, highlights clear differences in genetic structure between P. papatasi populations living the L. major-endemic area …." line 227-228".
  4. A comparison that will link between the galectin gene (47) and the CO1 haplotype from this article is needed.
  5. In Figure 2, it will be more powerful if you include an outstanding group.

Author Response

  1. The use of CO1 marker in sand flies is problematic, since it is one of most variable genetic markers that can be used in sand fly barcoding not only in Morocco. This article supports this assertion. The fact that there is a multiple haplotype in each focus weakens the finding, but on the other hand the results and the way the analysis is presented here are valued and important.

Nowadays the COI gene sequence is one of the most widely used genetic markers for making phylogenetic relationship. Here we use for phylogeographic purpose with aim to compare phlebotomine sandflies population genetic divergence in endemic and non-endemic areas. Therefore, its large variability, makes COI gene the ideal genetic marker for a such purpose.

  1. It is challenging to determine haplotype-based in one sequence only.

Yes, with single sequencing, error cannot be ruled out. Nevertheless, the manual editing of all the chromatogram has led to unambiguously define the mutation defining each haplotype.

  1. There is no other morphological, molecular or biological evidence to support the most important statement in this work "the CO1 gene, highlights clear differences in genetic structure between P. papatasi populations living the L. major-endemic area …." line 227-228".

We agree that additional marker must be used to more thoroughly define the population and their phylogenetic relationship. The sandfly populations we collected were indistinguishable by morphological observation and genomic analysis of the 18srRNA. These additional observations were added to the new version of our paper.

  1. A comparison that will link between the galectin gene (47) and the CO1 haplotype from this article is needed.

The sequencing of the galectin gene and COI analysis were not performed on the same samples. Additional information was added in the new version line 233-237 i.e: “This study pinpoints a polymorphism of the galectin gene (PpGalec) with a mutation affecting an amino acid involved in the substrate recognition. Although performed on a limited number of samples, this further point out an unequal distribution of this mutation in populations from endemic and nonendemic areas for zoonotic cutaneous leishmaniasis [48].”

  1. In Figure 2, it will be more powerful if you include an outstanding group.

We do not agree with the remark of the reviewer since our aim is to perform a phylogeographic analysis of P. papatasi populations. In the new version, additional analysis was performed and a phylogenetic tree with P. (Laroussius) ariasi as an outgroup was drawn. This demonstrates further that samples we sampled in Morocco are close to P. papatasi and clearly differ from COI sequences of the Laroussius or Transphlebotomus subgenus present in Morocco. Please see the new figure 2.

Reviewer 2 Report

The study presents data regarding the population genetic structure of P. papatasi from Morocco, the vector of Leishmania major pathogens. The conceptualisation is interesting comparing the genetic relationships of sand flies from Zoonotic Cutaneous Leishmaniasis endemic regions with sand flies from Zoonotic Cutaneous Leishmaniasis not endemic regions. However there are very important drawbacks in the way the results are presented as well as the analyses that are mentioned below.

First, the authors state in the abstract, as well as in lines 122-123 that “A 461-base pair fragment of the mtDNA cytochrome oxidase I (CO1) gene was amplified”, resulting in an alignment of 461 bp. However, they submitted sequences of 654 bp in length (GenBank Accession numbers: MT074053-MT74085). How is it possible to submit a sequence longer than your PCR product? Please explain

Also, the way that relationships among haplotypes are presented is confusing. They should have the same names everywhere, which is not the case in Figures 2 and 3. In Figure 2 they are PAP1, PAP2 PAP6, PAPA8 etc, whereas in Figure 3 they are H3, H4, H6 etc. Please correct

Since there is a large genetic differentiation observed between the two groups, Ouarzazate and Non-Ouarzazate group, an AMOVA would show it and support it better. This way the authors could show that the variation is among groups and not within groups. Although AMOVA is mentioned in the abstract, Materials and Methods and Discussion sections, I cannot find anywhere the AMOVA results. What percentage of variation was within groups and what among groups? Maybe present them in a Table, it is more important than Table 2 that represents information that can be found in GenBank or Table 4 that is a repetition of Table 3.

Although genetic differentiation is well discussed, genetic (haplotype) diversity of each population is not discussed at all. Since it is presented, it should be discussed in comparison with other Phlebotomus populations from previous studies. It can also provide useful information for each population analysed. 

I also suggest to discuss a little the suitability of COI as a DNA barcode according their findings. Did the authors detect any barcoding gap? Was there any case that the novel haplotypes could not assign correctly the the taxonomy of the collected specimens?

Some more detailed comments are following:

Abstract 

line 19: add the word “is” after “The disease”

line 21: replace “remaining” with “remain”

Introduction

line 69: The authors should add a sentence explaining this contradiction in genetic differentiation between studies based on sequencing of a single conserved gene and studies based on the highly polymorphic microsatellites. Could it be that cyt b and ITS2 were not capable to detect this differentiation?

line 78: I suggest to replace “natural” with “wild”. The term refers better to living organisms. Also change it in the rest manuscript.

Materials and methods

line 86: delete the comma after Morocco

lines 98-100: Since the authors did not design the primers for the needs of the present study, they have to provide a reference citing the study that designed the primers.

Results

lines 122-127: The authors state that the PCR amplification produced a band of approximately 500 bp, resulting in an alignment of 461 bp. However, the length of the submitted sequences is 654 bp. How is this possible???

line 136: Same as above: Here, in the subtitle of the Table 1, 668 bp of the COI gene are referred. Please define the correct length and correct it

line 156: I better suggest to include the original tree instead of the consensus, in order to be meaningful the length of the branches. You can indicate there the bootstrap values above 50%. This way it would also be more indicative of how closely related are the haplotypes of each group. 

lines 168-175: On the branches of the network, the number of variable sites (mutational steps) between each haplotype pair should be noted. Also, are there any median vectors found? If yes, it should be noted on the network that has to be reconstructed. In addition, not all Ouarzazate individuals fall into Ouarzazate group, thus the clustering is not so clear and this should be mentioned.

Discussion

line 210: Since a big part of the discussion and the general findings of the study refer to the Atlas mountains, I recommend to add a small description of the area, such as max height, environment etc, either here or in the Introduction

line 220: The authors have to decide to name the group with a stable term. Here, as well as in other parts of the manuscripts, it is named as Non-Ouarzazate while in Figure 2 it is mentioned as “No Ouarzazate”

lines 236-237 and 241-243: Please delete the guidelines for how to write the “Author Contributions” and “Acknowledgements” sections and keep only the parts referring to the present study.

Author Response

The study presents data regarding the population genetic structure of P. papatasi from Morocco, the vector of Leishmania major pathogens. The conceptualisation is interesting comparing the genetic relationships of sand flies from Zoonotic Cutaneous Leishmaniasis endemic regions with sand flies from Zoonotic Cutaneous Leishmaniasis not endemic regions. However there are very important drawbacks in the way the results are presented as well as the analyses that are mentioned below.

First, the authors state in the abstract, as well as in lines 122-123 that “A 461-base pair fragment of the mtDNA cytochrome oxidase I (CO1) gene was amplified”, resulting in an alignment of 461 bp. However, they submitted sequences of 654 bp in length (GenBank Accession numbers: MT074053-MT74085). How is it possible to submit a sequence longer than your PCR product? Please explain

Answer: We are sorry for the typo error, it is not a 461 -base pair but 661. Errors were corrected in the new version. If the original length of all the sequences was of about 661, their manual trimming resulted in a pool of good quality sequences of 653 bp. This precision was added in the text.

Also, the way that relationships among haplotypes are presented is confusing. They should have the same names everywhere, which is not the case in Figures 2 and 3. In Figure 2 they are PAP1, PAP2 PAP6, PAPA8 etc, whereas in Figure 3 they are H3, H4, H6 etc. Please correct

 Answer: We have clarified this point in the new version. In the figure 2 now appear, the code of the sample (PAP…) the haplotype (H..) the location (Labrouj, …) and the Genebank accession number.

Since there is a large genetic differentiation observed between the two groups, Ouarzazate and Non-Ouarzazate group, an AMOVA would show it and support it better. This way the authors could show that the variation is among groups and not within groups. Although AMOVA is mentioned in the abstract, Materials and Methods and Discussion sections, I cannot find anywhere the AMOVA results.

Answer: Results of AMOVA analysis has been added in the new version, in particular concerning the comparison between samples from endemic areas for ZCL as compare to non-endemic ones.

What percentage of variation was within groups and what among groups? Maybe present them in a Table, it is more important than Table 2 that represents information that can be found in GenBank or Table 4 that is a repetition of Table 3.

Answer: See the new version of our paper. We thank the reviewer for quoting this problem.

Although genetic differentiation is well discussed, genetic (haplotype) diversity of each population is not discussed at all. Since it is presented, it should be discussed in comparison with other Phlebotomus populations from previous studies. It can also provide useful information for each population analysed. 

I also suggest to discuss a little the suitability of COI as a DNA barcode according their findings. Did the authors detect any barcoding gap? Was there any case that the novel haplotypes could not assign correctly the the taxonomy of the collected specimens?

Answer: This is the first large study that use COI gene for population genetic analysis. We also find that the sequencing of this gene is of interest to sandfly taxonomy, see figure 2A. 

Some more detailed comments are following:

Abstract 

line 19: add the word “is” after “The disease”

Answer: Change done

line 21: replace “remaining” with “remain”

Answer: Change done

Introduction

line 69: The authors should add a sentence explaining this contradiction in genetic differentiation between studies based on sequencing of a single conserved gene and studies based on the highly polymorphic microsatellites. Could it be that cyt b and ITS2 were not capable to detect this differentiation?

Answer: We only review data on population genetic of sandfly, without making any assumption on the validity of these studies. We highlight that may be cyt b and ITS2 might not be useful to solve the degree of relatedness of individuals and groups. 

line 78: I suggest to replace “natural” with “wild”. The term refers better to living organisms. Also change it in the rest manuscript.

Answer: Change done

Materials and methods

line 86: delete the comma after Morocco

Done

lines 98-100: Since the authors did not design the primers for the needs of the present study, they have to provide a reference citing the study that designed the primers.

Answer: Reference from which the primers- set was picked has been added.

Results

lines 122-127: The authors state that the PCR amplification produced a band of approximately 500 bp, resulting in an alignment of 461 bp. However, the length of the submitted sequences is 654 bp. How is this possible???

Answer: No it is a typo error the correction was done

line 136: Same as above: Here, in the subtitle of the Table 1, 668 bp of the COI gene are referred. Please define the correct length and correct it

Answer: Sorry for this typo error the correction is done

line 156: I better suggest to include the original tree instead of the consensus, in order to be meaningful the length of the branches. You can indicate there the bootstrap values above 50%. This way it would also be more indicative of how closely related are the haplotypes of each group. 

Answer: See the modification perform on the figure 2, branch length was added etc…

lines 168-175: On the branches of the network, the number of variable sites (mutational steps) between each haplotype pair should be noted. Also, are there any median vectors found? If yes, it should be noted on the network that has to be reconstructed. In addition, not all Ouarzazate individuals fall into Ouarzazate group, thus the clustering is not so clear and this should be mentioned.

Answer: mutational steps were added as well as the median vectors. Non-Ouarzazate represent a mixed population and may encompass 2 populations named mix1 and mix 2 with on mixed population closer to the Ouarzazate group than the other.

Discussion

line 210: Since a big part of the discussion and the general findings of the study refer to the Atlas mountains, I recommend to add a small description of the area, such as max height, environment etc, either here or in the Introduction

A short description of the geography was added in the discussion section.

line 220: The authors have to decide to name the group with a stable term. Here, as well as in other parts of the manuscripts, it is named as Non-Ouarzazate while in Figure 2 it is mentioned as “No Ouarzazate”

Modification performed accordingly.

lines 236-237 and 241-243: Please delete the guidelines for how to write the “Author Contributions” and “Acknowledgements” sections and keep only the parts referring to the present study.

Correction done.

Reviewer 3 Report

The manuscript by Guernaoui et al. is a very interesting research paper dealing with the phylogeography of the sandfly, Phlebotomus papatasi, the main vector of the Old World cutaneous  leishmaniasis. The authors frame this work around the transmission of the Leishmania major from endemic regions towards non endemic regions. There was clearly a lot of work that went to this study, and I think this paper definitely falls within the scope of the journal. But I have a major issue with the analyses presented here i.e. the dataset itself. The sequences deposited by the authors in Genbank corresponding to the Genbank accession numbers mentioned in the paper are 654 bp long and most likely have been amplified using “ barcode primers” and certainly not with the LEP primer dataset as reported in the paper. It gives the impression that there are different datasets. When dealing with the dataset of longer sequences, we obtained 28 haplotypes , and not 33 haplotypes as reported in the paper with shorter sequences ( which is unlikely), the number of segregating sites is 32 , and not 37 as reported in the paper with shorter sequences , etc….I do not understand the rationale of using short CO1 sequences when longer sequences ( more informative) are available. Without a clear explanation of why the dataset deposited does not correspond to the dataset analyzed, I am concerned about the analyses presented in this paper, and therefore I do think the manuscript needs  a major revision pending upon authors' explanation as regards the dataset.

Author Response

The manuscript by Guernaoui et al. is a very interesting research paper dealing with the phylogeography of the sandfly, Phlebotomus papatasi, the main vector of the Old World cutaneous  leishmaniasis. The authors frame this work around the transmission of the Leishmania major from endemic regions towards non endemic regions. There was clearly a lot of work that went to this study, and I think this paper definitely falls within the scope of the journal. But I have a major issue with the analyses presented here i.e. the dataset itself. The sequences deposited by the authors in Genbank corresponding to the Genbank accession numbers mentioned in the paper are 654 bp long and most likely have been amplified using “ barcode primers” and certainly not with the LEP primer dataset as reported in the paper. It gives the impression that there are different datasets. When dealing with the dataset of longer sequences, we obtained 28 haplotypes in fact 27, and not 33 haplotypes as reported in the paper with shorter sequences ( which is unlikely), the number of segregating sites is 32 , and not 37 as reported in the paper with shorter sequences , etc….I do not understand the rationale of using short CO1 sequences when longer sequences ( more informative) are available. Without a clear explanation of why the dataset deposited does not correspond to the dataset analyzed, I am concerned about the analyses presented in this paper, and therefore I do think the manuscript needs a major revision pending upon authors' explanation as regards the dataset.

Answer: Firstly, we would like to thanks the reviewer for the critical review of our manuscript. All the comments raised by the were considered in this new version. All the analyses (phylogenetic, Network, AMOVA…) were performed with the dataset deposited in the genebank. We modified the haplotype number, network, and phylogenetic tree in accordance with the remarks. These new analyses do not change our conclusion on the genetic differentiation between P. papatasi populations from endemic and non-endemic areas for ZCL, it reinforces it!!! We are really grateful to the anonymous reviewer for raising serious concern on this manuscript. We hope that this new version of the manuscript will be now acceptable for publication.

Round 2

Reviewer 1 Report

 It is a pity, that the connection between the COI polymorphism was not made with galectin gene mutation. if this had been done the work would be much more significant. 

Author Response

Yes we agree with this comment and that has to be done with some new samples.

Reviewer 2 Report

I am satisfied with the improvements and corrections made by the authors and therefore I recommend publication of the article

Author Response

Thank you for having thoroughly reviewed the manuscript.

Reviewer 3 Report

This is a very interesting paper. The authors conclude that there is indeed a high genetic structure in the phylogeographic pattern in haplotypes of Phlebotomus papatasi in Morocco.

I have some concerns about this conclusion as the data and results as presented in the manuscript do not support this conclusion. One major ambiguity lies in the” reference groups “, endemic areas or Ouarzazate versus non endemic areas or non Ouarzazate. It is not clear at all when looking at Figure 2B where the non Ouarzazate group comprised haplotypes (H10, H11, H12 and H13) which are only present in Ouarzazate ! This is very confusing and I would recommend to clarify this.

The results as they are presented in the manuscript do not support the conclusion of the authors regarding the genetic structure in the phylogegraphic pattern. Therefore the discussion should be re-written accordingly

I have others comments listed hereafter

Page 1, line 25 : I only count 59 individuals, not 89.

Page 4, Data analysis section

  • Could you mention the software that you have been using to analyze the haplotype diversity (Hd)
  • When referring to the nucleotide diversity I would suggest to refer to π statistics only.

Page 4, Results section: the size of the police seems different

Line 88 Table 1 specify which locality (besides Ouarzazate) is in the endemic area of L. major

Line 117. The phylogenetic analysis as presented in Figure 2A refers to a different alignment ??? than the alignment presented for the intraspecific analysis ( line 123 to 126) as you added sequences of three congeners retrieved from Genbank and we could only assume that this alignment is 654 bp long. You should specify this.

Line 121 band is approximately 700bp

Line 122 sequences of 681 bp were manually trimmed

Line 123 write length

Line 126-127 between brackets delete the words “Accession numbers “

Accession should be MT074053-54; MT075056-57; MT075059-61; MT075064; MT075075; MT075078-85)

Line 130 provide the value of the haplotype diversity ± standard deviation. It seems to me (based on my calculation) the that the haplotype diversity is high and not low as it is written

Line 131 delete Thetas and Theta PI and replace by π value ± standard deviation

Line 133 Although the two values of Tajima’s D and Fu’s Fs are negative , they are not significant, which means that the polymorphism is random and not under pressure , therefore you have to be careful when drawing conclusion

Line 140 There are only 28 haplotypes in Table 2, not 33

Line 145 write thirty

Line 157 refer to figure 2A

Line 158 the two “clades” are absolutely not supported by your bootstrap values ( 50% is too low) ; I would suggest to remove figure 2B and keep 2A only .

Line 172 The median joining network is not highly structured at all ! the MIX2 is slightly differentiated from the others , that ‘s all.

In table 3, you are referring to 10 localities, therefore we should refer to these 10 localities in the network , not 14.

Line 186 you are referring to Fst, you should give the Fst values obtained between your different groups in order to support your conclusion . In that regards the line 195 is not clear . is the Fst of 0.31between mix 1 and 2 or Ouarzazate and the others ? please clarify

This is a very interesting paper. The authors conclude that there is indeed a high genetic structure in the phylogeographic pattern in haplotypes of Phlebotomus papatasi in Morocco.

I have some concerns about this conclusion as the data and results as presented in the manuscript do not support this conclusion. One major ambiguity lies in the” reference groups “, endemic areas or Ouarzazate versus non endemic areas or non Ouarzazate. It is not clear at all when looking at Figure 2B where the non Ouarzazate group comprised haplotypes (H10, H11, H12 and H13) which are only present in Ouarzazate ! This is very confusing and I would recommend to clarify this.

The results as they are presented in the manuscript do not support the conclusion of the authors regarding the genetic structure in the phylogegraphic pattern. Therefore the discussion should be re-written accordingly

I have others comments listed hereafter

Page 1, line 25 : I only count 59 individuals, not 89.

Page 4, Data analysis section

  • Could you mention the software that you have been using to analyze the haplotype diversity (Hd)
  • When referring to the nucleotide diversity I would suggest to refer to π statistics only.

Page 4, Results section: the size of the police seems different

Line 88 Table 1 specify which locality (besides Ouarzazate) is in the endemic area of L. major

Line 117. The phylogenetic analysis as presented in Figure 2A refers to a different alignment ??? than the alignment presented for the intraspecific analysis ( line 123 to 126) as you added sequences of three congeners retrieved from Genbank and we could only assume that this alignment is 654 bp long. You should specify this.

Line 121 band is approximately 700bp

Line 122 sequences of 681 bp were manually trimmed

Line 123 write length

Line 126-127 between brackets delete the words “Accession numbers “

Accession should be MT074053-54; MT075056-57; MT075059-61; MT075064; MT075075; MT075078-85)

Line 130 provide the value of the haplotype diversity ± standard deviation. It seems to me (based on my calculation) the that the haplotype diversity is high and not low as it is written

Line 131 delete Thetas and Theta PI and replace by π value ± standard deviation

Line 133 Although the two values of Tajima’s D and Fu’s Fs are negative , they are not significant, which means that the polymorphism is random and not under pressure , therefore you have to be careful when drawing conclusion

Line 140 There are only 28 haplotypes in Table 2, not 33

Line 145 write thirty

Line 157 refer to figure 2A

Line 158 the two “clades” are absolutely not supported by your bootstrap values ( 50% is too low) ; I would suggest to remove figure 2B and keep 2A only .

Line 172 The median joining network is not highly structured at all ! the MIX2 is slightly differentiated from the others , that ‘s all.

In table 3, you are referring to 10 localities, therefore we should refer to these 10 localities in the network , not 14.

Line 186 you are referring to Fst, you should give the Fst values obtained between your different groups in order to support your conclusion . In that regards the line 195 is not clear . is the Fst of 0.31between mix 1 and 2 or Ouarzazate and the others ? please clarify

Author Response

This is a very interesting paper. The authors conclude that there is indeed a high genetic structure in the phylogeographic pattern in haplotypes of Phlebotomus papatasi in Morocco.

I have some concerns about this conclusion as the data and results as presented in the manuscript do not support this conclusion. One major ambiguity lies in the” reference groups “, endemic areas or Ouarzazate versus non endemic areas or non Ouarzazate. It is not clear at all when looking at Figure 2B where the non Ouarzazate group comprised haplotypes (H10, H11, H12 and H13) which are only present in Ouarzazate ! This is very confusing and I would recommend to clarify this.

Answer: from the figure 2b that show cladistic analysis of P. papatasi haplotype suggest that a group of haplotype constitutes a cluster that comprises a majority of haplotypes detected in Ouarzazate (Ouarzazate group).  Nevertheless some haplotypes express some relatedness with haplotypes of the Non-ouarzazate group (H10, H11, H12). We agree that Boostrap values we get that are > to 50%, are to low to argue for clade definition. Nevertheless, the phylogenetic tree pinpoints that haplotypes, primarily detected in the Ouarzazate area, are somewhat closer to some haplotypes found in the Non-Ouarzazate areas. But that for a majority of haplotypes they are genetically closed and privative of the Ouarzazate area, as disclosed by the network.

The results as they are presented in the manuscript do not support the conclusion of the authors regarding the genetic structure in the phylogegraphic pattern. Therefore the discussion should be re-written accordingly

Answer: Our paper does not claim definitive and absolute conclusions on genetic differentiation between P. papatasi of the Ouarzazate areas and those of the Non-ouarzazate areas. The situation is, for sure, more complex!!! It highlights, that the genetic structure (haplotype composition) between these two areas is not equivalent and that, maybe, the physical barrier of the Atlas mountain has limited the diffusion of P. papatasi to the non-endemic areas for ZCL. This is our hypothesis, supported by our analysis of the sample we get. We modified the conclusions accordingly.

I have others comments listed hereafter

Page 1, line 25 : I only count 59 individuals, not 89.

Answer: correction done

Page 4, Data analysis section

  • Could you mention the software that you have been using to analyze the haplotype diversity (Hd)
  • When referring to the nucleotide diversity I would suggest to refer to π statistics only.

Answer: a supplementary data was added that contains a summary of the statistics computed with Arlequin and Pi statistics were added also in the text.

Page 4, Results section: the size of the police seems different

Answer: Corrected

Line 88 Table 1 specify which locality (besides Ouarzazate) is in the endemic area of L. major

Answer: Unfortunately no other localities than Ouarzazate.

Line 117. The phylogenetic analysis as presented in Figure 2A refers to a different alignment ??? than the alignment presented for the intraspecific analysis ( line 123 to 126) as you added sequences of three congeners retrieved from Genbank and we could only assume that this alignment is 654 bp long. You should specify this.

Answer: information added.

Line 121 band is approximately 700bp

Answer: correction done

Line 122 sequences of 681 bp were manually trimmed

Answer: corrected

Line 123 write length

Answer: corrected

Line 126-127 between brackets delete the words “Accession numbers “

Answer: corrected

Accession should be MT074053-54; MT075056-57; MT075059-61; MT075064; MT075075; MT075078-85)

Answer: corrected

Line 130 provide the value of the haplotype diversity ± standard deviation. It seems to me (based on my calculation) the that the haplotype diversity is high and not low as it is written

Answer: haplotype diversity added.

Line 131 delete Thetas and Theta PI and replace by π value ± standard deviation

Answer: corrected

Line 133 Although the two values of Tajima’s D and Fu’s Fs are negative , they are not significant, which means that the polymorphism is random and not under pressure , therefore you have to be careful when drawing conclusion

Line 140 There are only 28 haplotypes in Table 2, not 33

Answer: corrected

Line 145 write thirty

Answer: corrected

Line 157 refer to figure 2A

Answer: corrected

Line 158 the two “clades” are absolutely not supported by your bootstrap values ( 50% is too low) ; I would suggest to remove figure 2B and keep 2A only .

Answer: we propose to discuss on “groups” that encompass “individuals having related CO1 sequences” and not clades, since we bootstrap value are not as high to support the differentiation in clades.

Line 172 The median joining network is not highly structured at all ! the MIX2 is slightly differentiated from the others , that ‘s all.

Answer: The network is structured into 3 groups that are more or less differentiated. Nevertheless, we statistically analyzed Ouarzazate and Non-Ouarzazate (Mix 1+ Mix2).

In table 3, you are referring to 10 localities, therefore we should refer to these 10 localities in the network, not 14.

Answer: Error was corrected.

Line 186 you are referring to Fst, you should give the Fst values obtained between your different groups in order to support your conclusion. In that regards the line 195 is not clear. is the Fst of 0.31between mix 1 and 2 or Ouarzazate and the others ? please clarify

Answer: We test only Ouarzazate and non-Ouarzazate groups, the later including the Mix 1 and Mix 2 groups. The Fst refers to this test and is significant at P=0.05.

Round 3

Reviewer 3 Report

The authors have been taking into consideration most of the remarks made to the previous version. The  supplementary data 1 for statistics (Supp data S1) follows Arlequin's format and therefore is not suitable for publication. Please edit a table using word. 

Author Response

Results from Arlequin output formated under word format as requested.